# Global burden of fall and associated factors among individual with low vision: A systematic-review and meta-analysis

Kingsley Ekemiri[1]*, Chioma Ekemiri[2], Ngozika Ezinne[1], Victor Virginia[3], Osaze Okoendo[4], Robin Seemongal-Dass[5], Diane Van Staden[6], Carl Abraham[7]

1 Department of Optometry, Faculty of Medical Sciences, The University of the West Indies, St Augustine Campus, Tunapuna, Trinidad and Tobago, 2 Department of Health Promotion, Faculty of Medical Sciences, The University of the West Indies, St Augustine Campus, Tunapuna, Trinidad and Tobago, 3 School of Nursing, Faculty of Medical Sciences, The University of the West Indies, St Augustine Campus, Tunapuna, Trinidad and Tobago, 4 Department of Optometry, Courts Optical, Couva, Trinidad and Tobago, 5 Ophthalmology Unit, Faculty of Medical Sciences, The University of the West Indies, Tunapuna, Trinidad and Tobago, 6 Faculty of Health Sciences and Social Development, University of British Columbia, Tunapuna, South Africa, 7 Department of Optometry and Vision Sciences, University of the Cape Coast, Coast, Ghana

* iamekemiri@gmail.com

**Data Availability Statement:** All relevant data are within the manuscript and its Supporting Information files.

## Abstract

### Introduction

Low vision has a significant global health problem that impacts the personal, economical, psychological, and social life of an individual. Each year around 684 000 individuals die from falls, 80% of these deaths occur are in low- and middle-income countries. The risk of falling significantly increases with visual impairment. This review aimed to determine the global pooled prevalence of fall and associated factors among individuals with low vision.

### Methods and materials

Systematic search of published studies done on PubMed, EMBASE, MEDLINE, Cochrane, Scopus, Web of Science CINAHL and, Google Scholar. The Preferred Reporting Items for Systematic Reviews and Meta-Analyses guidelines were used to report the findings. Quality of studies was assessed using the modified Newcastle-Ottawa Scale (NOS). Meta-analysis was performed using a random-effects method using the STATA™ Version 14 software.

### Result

Thirty-five (35) studies from different regions involving 175,297 participants included in this meta-analysis. The overall pooled global prevalence fall among individual with low vision was 17.7% (95% CI: 16.4–18.9) whereas the highest prevalence was 35.5%; (95% CI: 28.4–42.5) in Australia and the lowest was 19.7%; (95% CI: 7.6–31.8) seen in South America. Fear of falling (OR: 0.16(95%CI 0.09–0.30), and severity of visual impairment (OR: 0.27 (95%CI (0.18–0.39) increases the odds of falling.

**Funding:** The author(s) received no specific funding for this work.

**Competing interests:** NO authors have competing interests The authors have declared that no competing interests exist.

## Conclusion

As one cause of accidental death, the prevalence of falls among individuals with low vision is high. Fear of falling and severity of falling increases the odds of falling. Different stake-holders should give due attention and plan effective strategies to reduce the fall among this population.

## Introduction

Low vision is the existence of a visual impairment that results in a disability or a vision loss that can't be corrected by medical or surgical treatments or conventional eyeglasses [1]. According to the World Health Organization (WHO), a person with low vision is a one who has impairment of visual functioning even after treatment and/or refractive correction, and has a visual acuity of less than 6/18 to light perception, or a visual field of less than 10˚ from the point of fixation, but who uses, or is potentially capable to use, vision for the planning and/or performing of activities [2].

According to different studies, the causes of low vision include: cataract, nystagmus, reti-nopathy, optic atrophy, glaucoma, refractive error, retinal disorders, albinism, and retinitis pigmentosa [3–5]. It has a significant global health problem that impacts the personal, eco-nomical, psychological, and social life of an individual [6, 7]. It also affects health -related qual-ity of life [8–11],.reduces functioning [12], and generally it has a substantial impact on activities of daily living [13]. In addition, the low vision adversely affects balance ability; increase mortality risks, and results in deterioration in physical functioning [14–16].

Worldwide, falls are the next prominent causes of unintended injury deaths. Each year an approximately 684,000 individuals die from falls, of which over 80% are in Low- and Middle-Income Countries (LMIC) [17]. Studies indicated that the risk of falling is significantly corre-lated with vision related impairment [18, 19].

Besides globally, 1.1 billion people are known living with vision loss, from these 43 million people are blind, 295 million people have moderate to severe visual impairment, 258 million people have mild visual impairment, and 510 million people have near vision problems [20]. The number of blind persons increased from 34.4 million in 1990 to 49.1 million in 2020 [21]. As its prevalence increases, the occurrence of fall accident will be also expected to rise.

Even though of the increased burden of low vision in different parts of the world, there are a number of barriers to access the low vision services: which include lack of awareness of ser-vices by people with low vision, many people do not identify themselves as having low vision [22], mental health problems, denial of need for low-vision aid, poor physical health, lack of transportation, lack of referrals, communication failure, misconceptions of the service, nega-tive societal views, influence of family and friends, insufficient visual impairment to warrant services, cost of the service, reduced perception of vision loss relative to other losses in life, and educational level [23, 24]. On the behalf of the health professionals, there is a lack of awareness about low vision services [25], and lack of awareness on referral criteria and available low vision care [26].

Vision loss and fall are interrelated, studies show that vision loss is high among those who fall, and vision loss may also be a contributing factor to falls [27, 28]. If an individual encoun-ters a fall, it results poor health and well-being, decreased activity of daily living, and social par-ticipation, lower life satisfaction, and it affects the quality of life of the individuals [29–32]. Identifying the various factors that may contribute to falls in this population, especially those

that may be modifiable will help in reducing the incidence of falling. So this systematic review and meta-analysis aimed to determine the global pooled prevalence of fall and associated factors among individuals with low vision.

## Objectives of the review

✓ To determine the pooled global prevalence of falls among individuals with low vision

✓ To identify the associated factors falls among individuals with low vision

## Methods and materials

### Inclusion and exclusion criteria

✓ The studies were included in the analysis if

✓ The study participants were individuals with low vision (visual acuity >0.3 logmar (6/12 Snellen)

✓ The prevalence of falls among individuals with low vision was reported.

✓ It reported on fall related injury within the past two years

✓ Reported fall were among adults.

✓ Both published and unpublished studies till December 2023 will be included in this analysis. Studies were excluded if

✓ Overall prevalence of fall in individual with low vision not report.

✓ It had greater study duration either prospectively or retrospectively.

✓ The prevalence of fall after corrective surgery.

### Information sources, search strategy, and study selection

Primarily, electronic searches were used to extract studies. PubMed, Scopus, Web of Science, JStore, and African journal online databases were used. The Cochrane acronym POCC (population, Condition, and Context) related key terms were used to retrieve studies in PubMed. The key terms used were, "Low vision OR visual impairment OR vision loss OR Ophthalmic conditions OR Low vision status OR declining vision OR vision impairment OR vision loss OR Visual field loss OR poor vision OR lowered vision OR fall OR "injury". Search limiters, such as study design and language of publication, were used. The identified articles were then exported into EndNote version 7.0 to remove duplicates. Also, manual and references searching was done to include studies not indexed in above mentioned sites. Finally the Preferred Reporting Items for Systematic Reviews and Meta-analyses (PRISMA) statement 2020 guidelines was used to report this study [33].

### Data collection process

After agreement with the search strategy, data extraction was done by all authors independently by using a data extraction format prepared in a Microsoft Excel 2013 spreadsheet; containing author's name, publication year, study design, sample size, setting, the prevalence of fall among individuals with low vision.and associated factors

## Data items

The primary outcome of this review is the global pooled prevalence of fall among individuals with low vision and the secondary outcome is the factors affecting fall among individuals with low vision. All variables of low vision or the causes of low vision were independently searched as outcome variable.

## Study risk of bias assessment /quality assessment of studies

The modified Newcastle-Ottawa Scale (NOS) for cross-sectional studies was used to assess the quality of studies, which has three components categorised as Selection, Comparatively, and outcome assessment methods, which scores out of 10. Studies that scored five or more on the NOS were included [34]. Any disagreement while assessing the quality of the study was resolved through careful examine of the studies together by all authors.

## Effect measures

The prevalence fall among individuals with low vision proportion were taken to measure the effect; whereas for the associated factors odds ratio was taken.

## Data analysis and synthesis methods

After the abstraction of all eligible studies; the data were exported to Stata software version 14 for analysis. For studies that reported the overall prevalence of fall in individuals with and without low vision, the proportion was calculated by considering of those individuals with low vision as denominator. A random-effects model was used considering the heterogeneity of studies. The study heterogeneity was assessed using Higgin's $I^2$ and Cochran's Q method. $I^2$ values of 25%, 50%, and 75% were considered low, moderate, and high heterogeneity, respectively. Subgroup analysis was also conducted by region. Also, funnel plot and Egger's test were used to check publication bias. Sensitivity analyses were conducted to assess robustness of the synthesized results.

# Results

## Study selection and characteristics

The search strategy retrieved 4,986 published original articles. After the removal of duplicate articles, 3785 articles remained. Following further screening, 984 articles were assessed for eligibility, of which 949 articles were excluded because they did not report the outcome of interest and did not fulfill the inclusion criteria. Finally, 35 studies were included in the analysis (Fig 1).

The 1 included [35–69] studies had 175,297 study participants with 31employing a cross-sectional design, 3 cohort, and 1 case control study design. The final sample size ranged from 48 [69]– 94,311 [49]. Most studies were conducted in North America, and the prevalence of fall ranges from 0.0176 [38] - 74 [60] (Table 1).

## Prevalence of fall among individuals with low vision

The overall pooled global prevalence fall among individuals with low vision was 24.1% (95% CI: 22.7, 25.5) with a heterogeneity index of 99.9% and P-value of $< 0.001$ (Fig 2) and because the Eggers test was found to be significant (P = 0.002), the final pooled prevalence was corrected for Duval and Tweedie's trim and fill analysis and was found to be 17.7 (95% CI: 16.4–18.9).

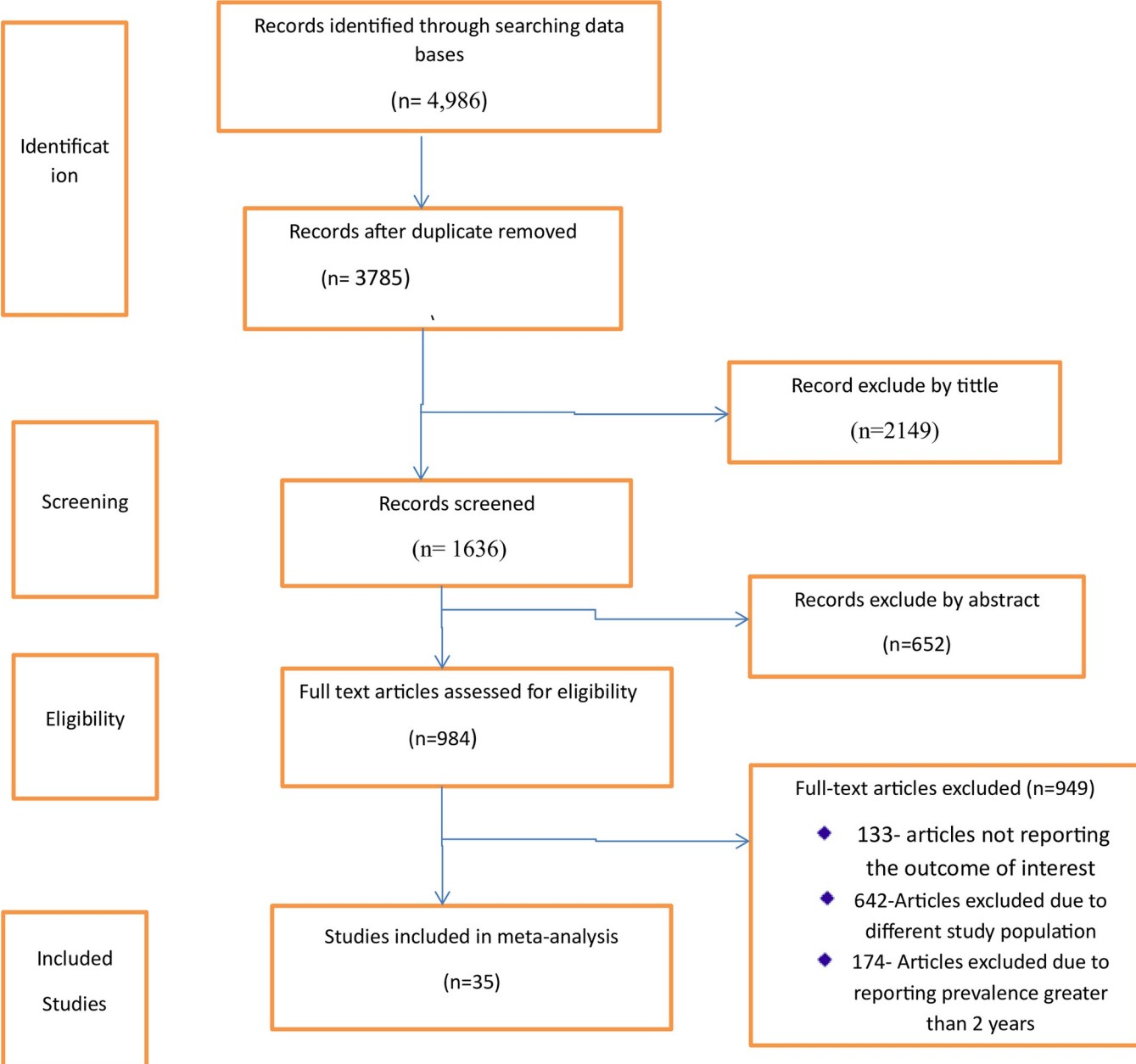

**Fig 1. PRISMA flow diagram of study selection for the global prevalence of fall among individual with low vision.**

## Meta-regression

Meta-regression was conducted using the year of publication and sample size as a covariate to identify the source of heterogeneity. It was indicated that there is no effect of publication year and sample size on heterogeneity between studies (Table 2).

## Subgroup analysis

Subgroup analyses revealed a marked variation across continents, with the highest prevalence 35.5%; (95% CI: 28.4–42.5), $I^2$ = 96.1%) in Australia and the lowest 19.7%; (95% CI: 7.6–31.8), $I^2$ = 100%) seen in South America (Fig 3).

**Table 1. Characteristics of the studies included in the systematic review and meta-analysis.**

| Authors Name | Publication Year | Study area | Study design | sample | Prevalence%(95%CI) |
|---|---|---|---|---|---|
| Gashaw M, et al | 2020 | Gondar | Cross -sectional | 328 | 26.8(22.0–31.5) |
| Mengste YL, et al | 2023 | Addis Ababa | Cross-sectional | 487 | 36.1(31.8–40.3) |
| Gashaw M, Adie Admass B. | 2021 | Gondar | Cross -sectional | 320 | 24.7(19.9–29.4) |
| KANG MJ, RIM TH, KIM SS. | 2016 | South Korea | Cross -sectional | 28899 | 0.0176(0.0023–0.032) |
| Lamoreux EL, et al | 2008 | Singapore | Cross -sectional | 1190 | 40.3(37.5–43.08) |
| Gupta P, et al | 2023 | Singapore | Cross -sectional | 1972 | 16.3(14.6–17.9) |
| Kuang TM, et al | 2008 | China | Cross -sectional | 1361 | 4.6(3.48–5.71) |
| Lamoureux E, et al | 2010 | Australia | Cross -sectional | 127 | 40(31.47–48.52) |
| Yip JL, et al | 2014 | UK | Cross- sectional | 8317 | 26.7(25.74–27.65) |
| Campagna G, et al | 2018 | USA | Cross -sectional | 809 | 7.4(5.59–9.20) |
| BOptom RQ, et al | 1998 | Australia | Cross -sectional | 3299 | 24.9(23.42–26.37) |
| Mehta J, et al | 2021 | UK | Cross -sectional | 585 | 16.4(13.39–19.40) |
| Freeman EE, et al | 2007 | USA | Cross -sectional | 2312 | 29(27.15–30.84) |
| Marmamula S, et al | 2020 | India | Cross -sectional | 1074 | 38(35.09–40.90) |
| Crews JE, et al | 2016 | USA | Cross -sectional | 94311 | 28.9(28.61–29.18) |
| Ouyang S, et al | 2022 | China | Cross -sectional | 251 | 56.9(50.77–63.02) |
| Kulmala J, et al | 2008 | Finland | Cross -sectional | 188 | 53.7(46.57–60.82) |
| Krishnaiah S, Ramanathan RV | 2018 | India | Cross -sectional | 382 | 13.8(10.34–17.25) |
| To KG, et al | 2014 | Vietnam | Cross -sectional | 413 | 13(9.75–16.24) |
| Coleman AL. et al | 2007 | USA | Cross -sectional | 4071 | 16(14.87–17.12) |
| Black AA, e al | 2008 | Australia | Cross -sectional | 65 | 35(23.40–46.59) |
| Bhorade AM, et al | 2021 | USA | Cross -sectional | 138 | 36(27.99–44.0) |
| Black AA, et al | 2011 | Australia | Cross -sectional | 71 | 44(32.45–55.54) |
| Baig S, et al | 2016 | USA | Cross -sectional | 116 | 25(17.11–32.88) |
| Patino CM, et al | 2010 | USA | Cross- sectional | 3203 | 19(17.64–20.35) |
| Wood JM, et al | 2011 | Australia | Cross- sectional | 76 | 74(64.13–83.86) |
| Moghadam AN, et al | 2015 | Iran | Case-control | 48 | 22.9(11.01–34.78) |
| Pattaramongkolrit S, et al | 2013 | Thailand | Cross- sectional | 278 | 37.8(32.10–43.49) |
| McCarty CA, et al | 2002 | Australia | Cross- sectional | 2343 | 20(18.38–21.61) |
| Kwan MM, et al | 2012 | Taiwan | Cross- sectional | 260 | 44.5(38.45–50.54) |
| Rosenblatt TR, et al | 2023 | USA | Cohort | 13385 | 0.063(0.020–0.105) |
| Glynn RJ, et al | 1991 | USA | Cross- sectional | 489 | 9.6(6.98–12.21) |
| Ehrlich JR, et al | 2019 | USA | Cross- sectional | 3933 | 27.6(26.20–28.99) |
| Lord SR, Dayhew J. | 2001 | Australia | Cohort | 148 | 21.7(15.05–28.34) |
| Haymes SA, et al | 2007 | Canada | Cohort | 48 | 35(21.50–48.49) |

## Publication bias

The publication bias was statistically tested by the Egger test (P = 0.002); which is significant. The funnel plot indicated publication bias as the graph appeared asymmetrical (Fig 4). After adjusting for publication bias by trim and fill analysis, the funnel plot appeared to be symmetrical (Fig 5).

## Sensitivity analysis

Additionally, in this systematic review and meta-analysis, sensitivity analysis was performed to determine how various sources of uncertainty contribute to the overall uncertainty among the studies, but the results indicated that uncertainty has an insignificant influence on pooled prevalence (Fig 6).

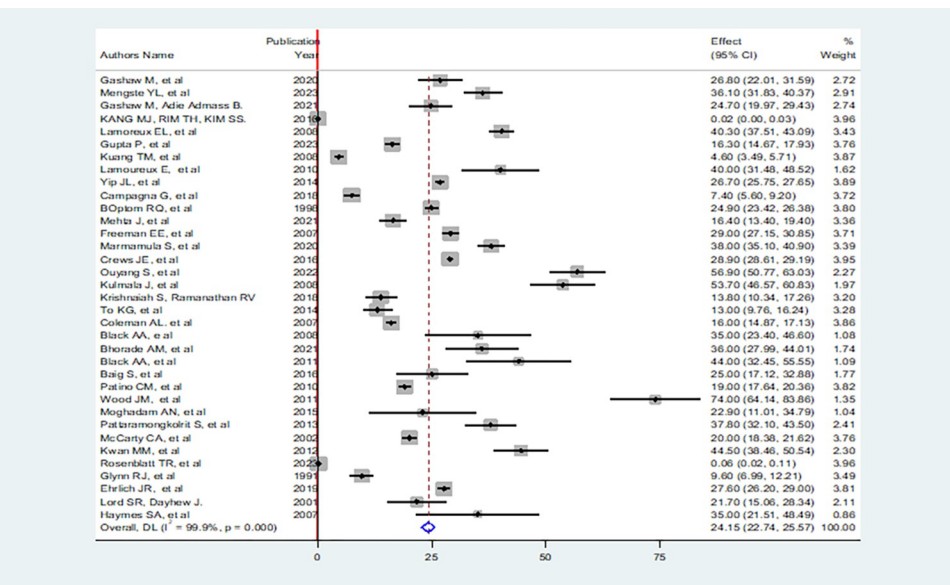

**Fig 2. Forest plot showing the pooled the global prevalence of fall among individual with low vision.**

## Factor associated factors fall among individual with low vision

Five variables were extracted to identify factors associated with fall among individuals with low vision. Of those, two variables (Fear of falling and Severity of visual impairment) were identified as significant factors (Table 3).

## Discussion

In this systematic review and meta- analysis, fall burden and associated factors was assessed among individuals with low vision. Different studies have shown that fall is a prevalent public health problem in the older population; in addition to advancement of their age, the presences of chronic diseases such as visual impairment worsens the problem in this group of individual [18, 70–74].

Globally, the estimated 10 million people who had visual impairment in 2015 are, estimated to be 55.7 million people by 2050 [75]; those patients with visual impairment at higher risk of falling [76].This shows that while the prevalence increases, those individuals are at high risk of health- related problems including falls.

According to our study, the overall pooled global prevalence falls among individuals with low vision was 17.7% (95% CI: 16.4–18.9); which is lower than pooled global prevalence of falls in the older adults 26.5% [77], this might be those individual with low vision take care to not fall knowing that they are at risk of falling compared to other individuals without visual impairment.

In addition, the subgroup analyses showed, the highest prevalence 35.5%; (95% CI: 28.4–42.5) in Australia and the lowest 19.7%; (95% CI: 7.6–31.8) seen in South America. This

**Table 2. Meta-regression analysis of factors affecting the between-study heterogeneity of depression.**

| Heterogeneity source | Coefficients | Std. Err. | P-value |
|---|---|---|---|
| Publication year | -0.2329166 | 0.2990457 | 0.442 |
| Sample size | -0.0003109 | 0.0001802 | 0.094 |

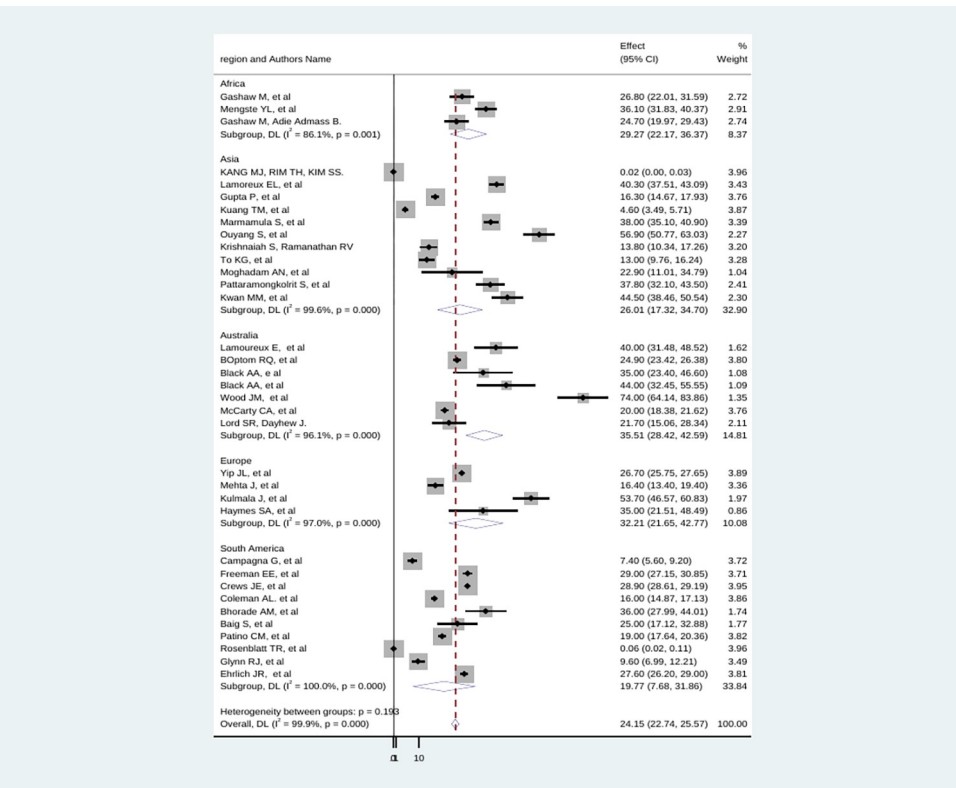

**Fig 3. Subgroup analysis of the global prevalence of fall among individual with low vision.**

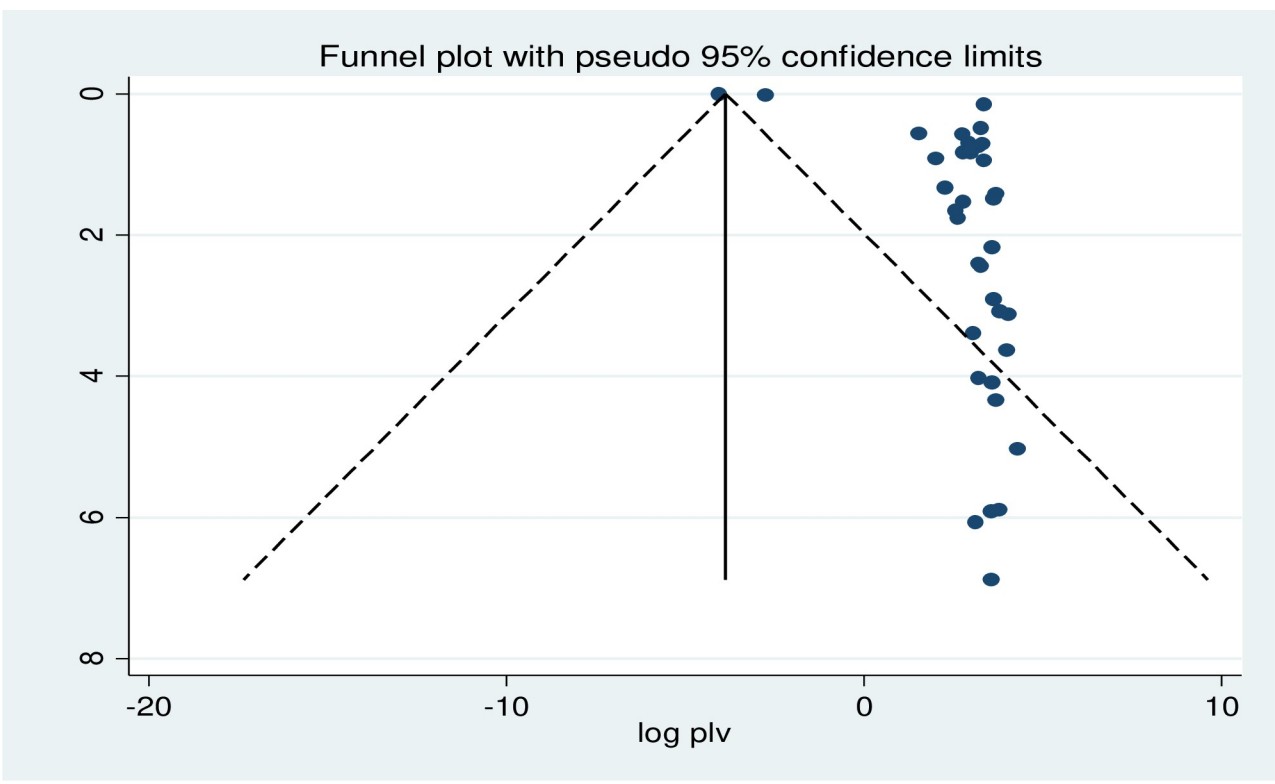

**Fig 4. Funnel plot to test publication bias in 35 studies with 95% confidence limits.**

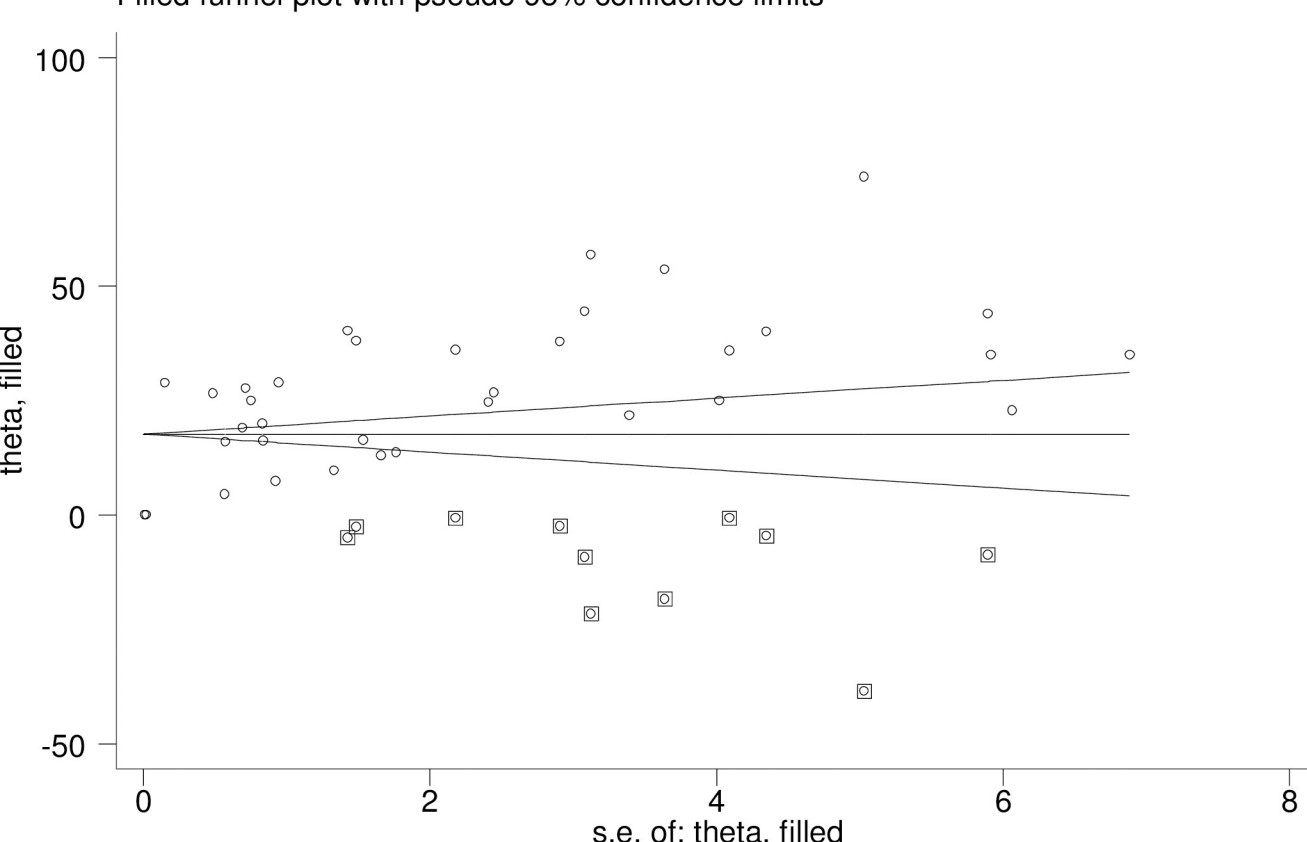

**Fig 5. Funnel plot after adjusting the trim and fill analysis.**

discrepancy might be due to evidence showing that the burden related to visual impairment and socioeconomic indicators were closely associated; as a result the occurrence of falls increased [78, 79].

In this study, falling was associated with fear of fall. Accordingly, those individuals who are concerned about fear of falling were 88.4% less likely to fall (OR: 0.16(95%CI 0.09–0.30), than those not concerned about the fear of falling. A different study also showed that fear of falling

**Table 3. Factor associated with fall among individuals with low vision.**

| Determinants | Comparison | No of studies | Sample size | OR(95%CI) | P–value | I² (%) | Heterogeneity test (p value) |
|---|---|---|---|---|---|---|---|
| Depression | Yes vs No | 2 | 648 | 0.98 (0.081–12.07) | 0.992 | 95.8 | < 0.001 |
| Gender | Men vs Women | 4 | 4354 | 0.87 (0.51–1.48) | 0.620 | 85.2 | < 0.001 |
| Taking of medication | Yes vs No | 2 | 648 | 0.991 (0.02–41.42) | 0.996 | 98.9 | < 0.001 |
| Fear of falling | Concerned vs Not concerned at all | 2 | 818 | 0.16 (0.09–0.30) | < 0.001 | 52.1 | 0.148 |
| Severity of visual impairment | Mild vs moderate and severe | 2 | 807 | 0.27 (0.18–0.39) | < 0.001 | 3.2 | 0.309 |

Accordingly, individuals who were concerned about the fear of falling were 88.4% less likely to fall (OR: 0.16(95%CI 0.09–0.30), P<0.001, I²: 52.1%, the heterogeneity test (P = 0.148) than an those not concerned about the fear of falling

Individuals with mild vision impairment 73% less likely to fall (OR: 0.27(95%CI (0.18–0.39)), P<0.001, I²: 3.2%, the heterogeneity test (p = 0.309) than those with moderate and sever vision impairment.

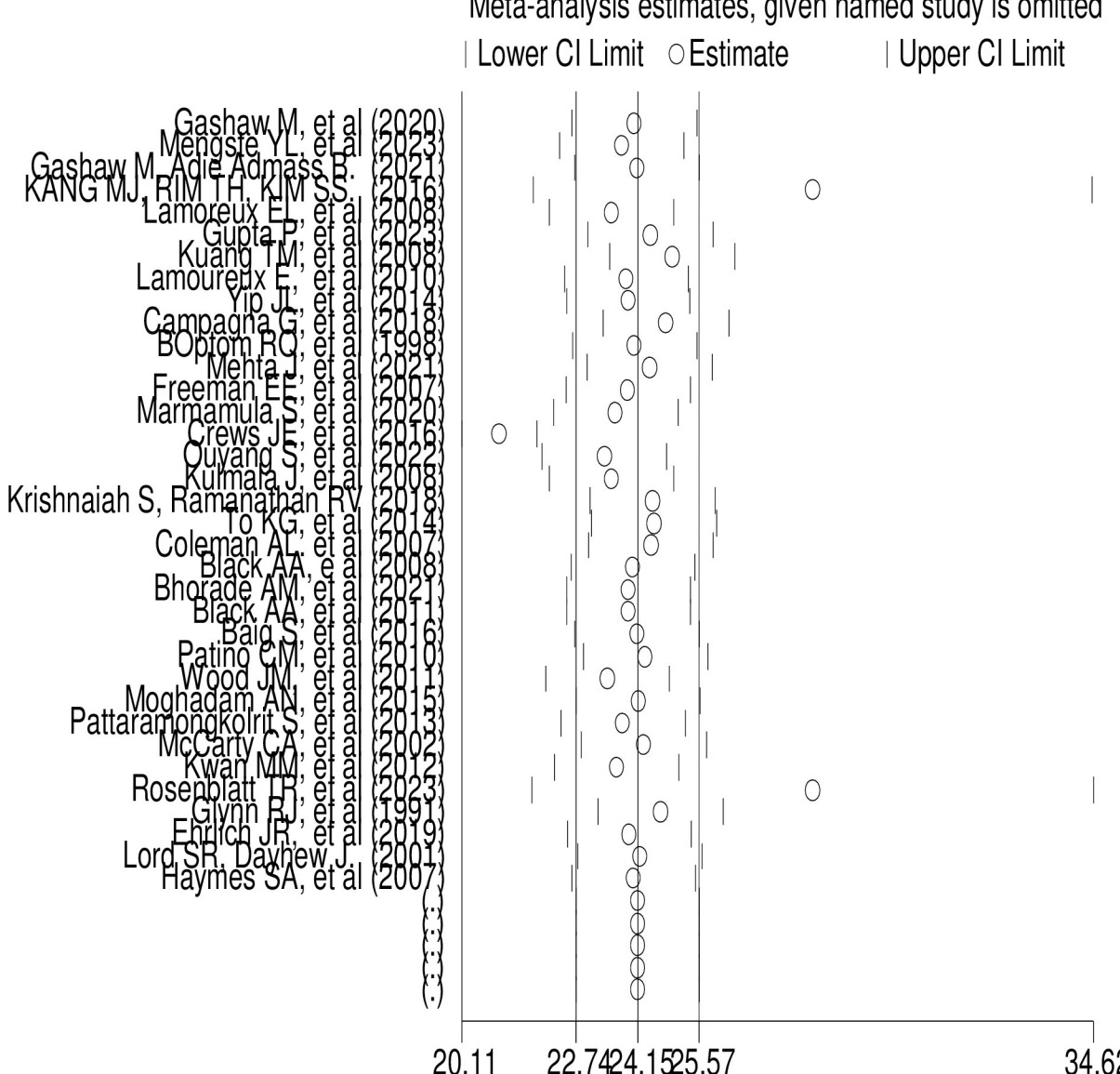

**Fig 6. Sensitivity analysis of pooled studies.** Sensitivity analysis of pooled the global prevalence of fall among individual with low vision for each study being removed one at a time.

was associated with the occurrence of falls and that the occurrence of falls might contribute to the development of fear of falling [80, 81].

The degree of vision impairment is associated with fall occurrence. In this study, individuals with mild vision impairment 73% less likely to fall (OR: 0.27(95%CI (0.18–0.39) than those with moderate and sever vision impairment. Studies show that the occurrence of repeated falls depends on the level of visual impairment [19, 82, 83].

## Conclusion

As one cause for accidental death, the prevalence of fall among individual with low vision is high. Fear of falling and severity of falling increases the odds of falling. Different

stakeholders should give due attention and plan effective strategies to reduce the fall among this population.

## Limitations of the study

This systematic review and meta-analysis presented the global prevalence of falls among individuals with low vision; however it might have faced the following limitations. First, the lack of studies from South America and Antarctica, may affect the generalizability of the findings to the world. Secondly, due to presence of significant heterogeneity and publication bias, the result should be interpreted cautiously. Finally, we faced difficulties in in comparing our findings due to the lack of regional and worldwide systematic reviews and meta-analysis.

## Supporting information

**S1 Checklist. PRISMA 2020 checklist.**
(DOCX)

**S1 Data.**
(XLSX)

## Acknowledgments

We would like to thank all authors of studies included in this systematic review and meta-analysis.

## Author Contributions

**Conceptualization:** Kingsley Ekemiri, Chioma Ekemiri.

**Data curation:** Kingsley Ekemiri, Victor Virginia, Osaze Okoendo, Carl Abraham.

**Formal analysis:** Osaze Okoendo, Diane Van Staden, Carl Abraham.

**Investigation:** Kingsley Ekemiri, Chioma Ekemiri, Victor Virginia, Osaze Okoendo.

**Methodology:** Kingsley Ekemiri, Chioma Ekemiri, Victor Virginia, Diane Van Staden, Carl Abraham.

**Project administration:** Ngozika Ezinne, Osaze Okoendo, Robin Seemongal-Dass, Carl Abraham.

**Resources:** Ngozika Ezinne, Osaze Okoendo, Robin Seemongal-Dass, Carl Abraham.

**Software:** Victor Virginia.

**Supervision:** Diane Van Staden.

**Validation:** Ngozika Ezinne, Victor Virginia, Robin Seemongal-Dass, Diane Van Staden, Carl Abraham.

**Visualization:** Chioma Ekemiri, Ngozika Ezinne, Osaze Okoendo, Robin Seemongal-Dass.

**Writing – original draft:** Kingsley Ekemiri, Chioma Ekemiri.

**Writing – review & editing:** Ngozika Ezinne, Victor Virginia, Robin Seemongal-Dass, Diane Van Staden.

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
