## [Decision Letter · Decision Letter 0]

7 Feb 2024

PONE-D-23-44176Global Burden of Fall Among Individual with Low Vision: A Systematic-Review and Meta-AnalysisPLOS ONE

Dear Dr. Ekemiri,

Thank you for submitting your manuscript to PLOS ONE. After careful consideration, we feel that it has merit but does not fully meet PLOS ONE’s publication criteria as it currently stands. Therefore, we invite you to submit a revised version of the manuscript that addresses the points raised during the review process.

We look forward to receiving your revised manuscript.

Kind regards,

Yuan-Pang Wang, M.D., Ph.D.

Academic Editor

PLOS ONE

Journal Requirements:

https://www.emerald.com/insight/content/doi/10.1108/IJPCC-09-2015-0033/full/html

https://pubmed.ncbi.nlm.nih.gov/24314403/

https://www.tandfonline.com/doi/full/10.1080/23311932.2019.1642981

https://downloads.hindawi.com/journals/ijcd/2021/6708865.pdf

In your revision ensure you cite all your sources (including your own works), and quote or rephrase any duplicated text outside the methods section. Further consideration is dependent on these concerns being addressed.

Additional Editor Comments:

Both reviewers have provided comments and suggested corrections to this manuscript. Please point to point to address their concerns.

Reviewers' comments:

Reviewer's Responses to Questions

**Comments to the Author**

1. Is the manuscript technically sound, and do the data support the conclusions?

Reviewer #1: Yes

Reviewer #2: Partly

2. Has the statistical analysis been performed appropriately and rigorously? 

Reviewer #1: Yes

Reviewer #2: Yes

3. Have the authors made all data underlying the findings in their manuscript fully available?

Reviewer #1: Yes

Reviewer #2: No

4. Is the manuscript presented in an intelligible fashion and written in standard English?

Reviewer #1: No

Reviewer #2: Yes

5. Review Comments to the Author

Reviewer #1: I have already commented it to rewrite it in standard English. I hope the authors will act accordingly. the manuscript technically sounded, the statistical analysis performed appropriately and rigorously.

Reviewer #2: it is an important work but needs revision

6. PLOS authors have the option to publish the peer review history of their article (what does this mean?). If published, this will include your full peer review and any attached files.

Reviewer #1: **Yes: **Mistire Teshome Guta

Reviewer #2: No

---

## [Author Response · Author response to Decision Letter 0]

1 Mar 2024

Dear Editor,

A rebuttal letter was sent and attached as "Response to Reviewers."

A marked-up copy was sent and labelled as Revised Manuscript with Track Changes

An unmarked version of revised paper was sent and labelled as "Manuscript"

The overlap was revised especially the introduction and Method

Response for reviewer 1

Dear reviewer thank you for your constructive comments and suggestion, we tried to incorporate all issues raised by you.

Manuscript PONE-D-23-44176 Global Burden of Fall Among Individual with Low Vision: A Systematic-Review and Meta-Analysis

Comments 

The tittle “Global Burden of Fall among Individual with Low Vision: A Systematic-Review and Meta-Analysis” is important because, the Vision loss and fall are interrelated, vision loss is high among those who fall, and vision loss may be a contributing factor to falls. Falling results poor health and well-being, decreased activity of daily living and social participation, lower life satisfaction, as well as it affects the quality of life of the individuals. 

I like to give minor comments and questions for this review.

1. I need to see the searching terms that you use for each data bases with their results? How do you get an access for each data bases other than PubMed?

Response- as we mentioned in method part we used those search terms for all databases after we extracted MeSH for our study based on POCC.

• We got all most all studies from PubMed, from other sites we only get 2 studies ( 1 Scopus indexed, so we accessed through institutional login.) the other from Google scholar.

 Regarding the access issue most of the health journals are indexed in PubMed in addition to that other sites, so we haven’t faced any challenges. But what we have done for those not indexed on those sited is we did manual searches.

2. The whole article has language and grammar problems.

Response- modified 

3. In line 129 it says “the keywords used were, Low vision OR visual impairment OR…..” are this key words or search terms?

Response—corrected based on your suggestion 

4. Check for the right citation: for example for citation 35………

Response—corrected 

5. Make the PRISMA diagram aesthetic.

Response---corrected 

6. Since you are using Preferred Reporting Items for Systematic Reviews and Meta-analyses (PRISMA)….. use its correct format. Like 1. Introduction, 2. Material and methods, 2.1. Study design and search strategy, 2.2. Study selection and eligibility criteria, 2.3. Study selection and quality appraisa, 2.4. Data extraction, 2.5. Measure, 2 .6. Statistical analysis, 2. 7. Publication bias, 3. Results………….. 4. Discussion, 4.1. Implications, 5. Conclusions, Financial disclosures, Declaration of Competing Interest, Acknowledgements, Appendix A. Supplementary data, References.

Response- we followed PRISMA 2020, so corrected based on your recommendation 

7. In line 235 make it 88.4%, it says 884%.

Response – corrected 

8. If the Egger’s test is found to be significant, it indicates the presence of publication bias in the meta-analysis. To correct the final pooled prevalence, one can use the trim-and-fill method1. This method involves trimming the studies that are causing funnel plot asymmetry and then filling the funnel plot with hypothetical studies to make it symmetrical. The final pooled prevalence is then calculated using the filled funnel plot. 

Response – mentioned on line 183 and 184

Response for reviewer 2

Dear reviewer thank you for your comments, we response for all comments raised by you.

Global Burden of fall and Associated factors among Individual with Low 2 Vision: A Systematic-Review and Meta-Analysis; It is an important piece of work but needs minor revision

Comments to author

The title on the system and word document is different , should be similar Global Burden of Fall Among Individual with Low Vision: A Systematic-Review and Meta-Analysis and Global Burden of Fall and associated factors Among Individual with Low Vision: A Systematic-Review and Meta-Analysis 

Response – corrected 

On page 1 line number 5 , Kingsley Ekemiri 1* 3 ,Chioma Ekemiri², Ngozika

Ezinne¹,Victor Virginia³,Osaze 4 Okoendo⁴ , Robin Seemongal-Dass⁵ , Diane Van

Staden⁶ , Carl Halladay Abraham⁷ , Low Vision Study Group¹

“ Low Vision Study Group” should be removed because it not the name of author

Response – removed 

The Corresponding Author is already showed by astrix and the affiliation so don’t need to be restated in title page

 Dr Kingsley Ekemiri(OD,MPH)

 Department of Optometry, Faculty of Medical Sciences, The University of West indies, St Augustine Campus, Trinidad and Tobago

Response – removed 

Under inclusion criteria- why authors only included reported fall injury within past two years?

Response – the author included studies which reported within two years is in order to minimize recall bias.

All included studies are published studies, why author dealt with unpublished study?

Response – corrected and we removed the word unpublished from main document, we don’t have unpublished studies

Under conclusion- author recommended …to give attention, do you think there is no attention for those groups?

Response – the current facts says fall is not given attention for those with low vision rather most literature suggests attention for older age groups

The contents of declaration letter is not correct, it is cover letter , please amend it

See plos one submission guideline at https://journals.plos.org/plosone/s/submission-guidelines

Response – corrected and removed from the manuscript 

All supporting information is not available

Response – we will make available/ attach as supporting information

---

## [Decision Letter · Decision Letter 1]

4 Apr 2024

Global Burden of Fall Among Individual with Low Vision: A Systematic-Review and Meta-Analysis

PONE-D-23-44176R1

Dear Dr. Ekemiri,

We’re pleased to inform you that your manuscript has been judged scientifically suitable for publication and will be formally accepted for publication once it meets all outstanding technical requirements.

Kind regards,

Yuan-Pang Wang, M.D., Ph.D.

Academic Editor

PLOS ONE

Additional Editor Comments (optional):

Reviewers' comments:

Reviewer's Responses to Questions

**Comments to the Author**

1. If the authors have adequately addressed your comments raised in a previous round of review and you feel that this manuscript is now acceptable for publication, you may indicate that here to bypass the “Comments to the Author” section, enter your conflict of interest statement in the “Confidential to Editor” section, and submit your "Accept" recommendation.

Reviewer #1: All comments have been addressed

Reviewer #2: All comments have been addressed

2. Is the manuscript technically sound, and do the data support the conclusions?

Reviewer #1: Yes

Reviewer #2: Yes

3. Has the statistical analysis been performed appropriately and rigorously? 

Reviewer #1: Yes

Reviewer #2: Yes

4. Have the authors made all data underlying the findings in their manuscript fully available?

Reviewer #1: Yes

Reviewer #2: Yes

5. Is the manuscript presented in an intelligible fashion and written in standard English?

Reviewer #1: Yes

Reviewer #2: Yes

6. Review Comments to the Author

Reviewer #1: The manuscript technically sound, statistical analysis been performed appropriately and rigorously and written in standard English. The authors well addressed the comments I send to them. So, I have no more comments.

Reviewer #2: the authors corrected all my comments and worthy if it is published by journal. I don't have additional comments

7. PLOS authors have the option to publish the peer review history of their article (what does this mean?). If published, this will include your full peer review and any attached files.

Reviewer #1: **Yes: **Mistire Teshome Guta

Reviewer #2: No

---

## [Editor Report · Acceptance letter]

16 May 2024

PONE-D-23-44176R1 

PLOS ONE

Dear Dr. Ekemiri, 

I'm pleased to inform you that your manuscript has been deemed suitable for publication in PLOS ONE. Congratulations! Your manuscript is now being handed over to our production team.

Kind regards, 

on behalf of

Dr. Yuan-Pang Wang 

Academic Editor

PLOS ONE